# Targeting NRF2 and FSP1 to Overcome Ferroptosis Resistance in *TSC2*-Deficient and Cancer Cells

**DOI:** 10.3390/cancers17162714

**Published:** 2025-08-21

**Authors:** Tasmia Tahsin, Darius K. McPhail, Jesse D. Champion, Mohammad A. M. Alzahrani, Madeleine L. Hilditch, Alexandre Faris-Orr, Brian L. Calver, James G. Cronin, Juan C. Mareque-Rivas, Darren W. Sexton, Stephen Fôn Hughes, Robert Steven Conlan, David Mark Davies, Andrew R. Tee

**Affiliations:** 1Institute of Life Science, Swansea University Medical School, Faculty of Medicine, Health & Life Science, Swansea University, Swansea SA2 8PP, UK; tasmiatahsin@gmail.com (T.T.); j.cronin@swansea.ac.uk (J.G.C.); r.s.conlan@swansea.ac.uk (R.S.C.); 2Maelor Academic Unit of Medical & Surgical Sciences (MAUMSS), Betsi Cadwaladr University Health Board (BCUHB), Faculty of Social & Life Sciences, Wrexham University, Gwenfro Buildings, Technology Park, Wrexham LL13 7YP, UK; stephen.hughes6@wales.nhs.uk; 3Division of Cancer and Genetics, Cardiff University, Heath Park, Cardiff CF14 4XN, UK; mcphaild@cardiff.ac.uk (D.K.M.); championjd1@cardiff.ac.uk (J.D.C.); zah10000@gmail.com (M.A.M.A.); hilditchm@cardiff.ac.uk (M.L.H.); abf-o@hotmail.fr (A.F.-O.); calverbl@cardiff.ac.uk (B.L.C.); daviesdm15@cardiff.ac.uk (D.M.D.); 4Department of Chemistry, Swansea University, Singleton Park, Swansea SA2 8PP, UK; juan.mareque-rivas@swansea.ac.uk; 5School of Pharmacy and Biomolecular Sciences, Liverpool John Moores University, Liverpool L3 3AF, UK; d.w.sexton@ljmu.ac.uk

**Keywords:** ferroptosis, tuberous sclerosis complex, mTOR, cancer resistance, NRF2, FSP1, RSL3

## Abstract

Cancer treatments often aim to induce cell death, but many tumors develop resistance to conventional therapies such as chemotherapy. Ferroptosis is a distinct, iron-dependent form of cell death that offers a promising alternative for overcoming this resistance. In this study, we investigated how cancer cells resist ferroptosis, focusing on models with elevated iron metabolism. Using Tuberous Sclerosis Complex 2 (TSC2)-deficient cells, as well as ovarian and breast cancer cell lines, we found that these cells were resistant to ferroptosis. However, this resistance could be reversed by inhibiting key antioxidant proteins. Our study points to new therapeutic strategies that target ferroptosis resistance mechanisms, offering potential treatments for otherwise difficult-to-treat cancers.

## 1. Introduction

Most cancer treatments aim to trigger cell death by either disrupting cell survival and growth pathways or by enhancing immune system recognition of tumors. Typically, conventional therapies result in apoptosis, a controlled and highly regulated form of cell death characterised by cell shrinkage, membrane budding, and phagocytosis by surrounding cells (see review [1]). However, many cancers develop resistance to therapies by disabling apoptotic pathways, necessitating alternative cytotoxic strategies. Given the limitations of apoptosis-inducing therapies, ferroptosis represents a promising alternative for targeting therapy-resistant tumors.

Ferroptosis is an iron-dependent form of regulated cell death that is driven by lipid peroxidation (LPO) and oxidative damage to cell membranes [2]. Unlike apoptosis, which relies on caspase activation, ferroptosis is primarily regulated by metabolic and oxidative stress pathways, with glutathione peroxidase 4 (GPX4) and ferroptosis suppressor protein 1 (FSP1) playing central roles in preventing LPO. While phospholipid peroxides are the driving force behind ferroptosis, the mechanisms of ferroptosis resistance are poorly understood. Morphologically, ferroptosis is distinct from apoptosis and necrosis. Cells undergoing ferroptosis exhibit shrinking mitochondria, disrupted mitochondrial cristae and increased cell membrane density [2].

Iron metabolism plays a crucial role in rapidly proliferating cells, supporting processes such as DNA/RNA synthesis, redox homeostasis, and ATP production. However, dysregulation of iron homeostasis and an increase in the labile iron pool can promote oxidative stress and sensitize cells to ferroptosis [2]. To maintain homeostasis and to prevent ferroptosis, cells must carefully regulate intracellular iron levels by modulating iron import, export, and storage in iron-sulfur clusters and ferritin. Additionally, redox defense mechanisms exist to combat ferroptosis. Perturbations in amino acid availability, particularly cysteine, play an important role due to its significance in the synthesis of glutathione (GSH), the most abundant intracellular antioxidant. GSH serves as a reducing agent to GPX4, a selenium-dependent enzyme, to detoxify lipid peroxides and prevent the accumulation of LPO in cell membranes [3]. Ferroptosis induction is typically accomplished by direct or indirect inhibition of GPX4, i.e., with Ras-selective lethal 3 (RSL3) treatment that directly inhibits GPX4 or with FIN56 treatment that promotes GPX4 degradation [4]. Ferroptosis can also be triggered by inhibiting the regeneration of the cellular antioxidant GSH via blockade of the cysteine-glutamate transporter Xc(−) system with erastin. Both inhibition of cysteine amino acid uptake and GPX4 are known to exacerbate LPO [5].

The transcription factor, nuclear factor E2-related factor 2 (NRF2/*NFE2L2*), defends against ferroptosis by upregulating the expression of antioxidant and cytoprotective genes that counteract the accumulation of toxic reactive peroxides [6]. Key downstream gene targets of NRF2 involved in ferroptosis are *GPX4*, *SLC7A11* (solute carrier family 7 member 11, a cysteine-glutamate antiporter), *HMOX1* (heme oxygenase 1) and *AIFM2* (AIF family member 2, which is the gene encoding FSP1). HMOX1 is involved in heme catabolism and is a pro-ferroptosis enzyme [7]. Increased iron levels in the cell, by autophagic degradation of the iron storage protein ferritin and the upregulation of HMOX1, further contribute to reactive oxygen species (ROS) production and ferroptosis induction. FSP1 functions as a NAD(P)H-dependent oxidoreductase that regenerates ubiquinol, thus preventing lipid oxidative damage and ferroptosis (reviewed in [4]).

To support their own growth, cancers often upregulate iron-linked metabolic pathways, increase iron uptake and can often develop a dependence on iron to become “iron addicted” (reviewed in [8]). Simultaneously, cancers often upregulate redox defense mechanisms that counteract ferroptosis, thus conferring ferroptosis resistance. Ferroptosis resistance protects cancer cells from iron-induced cell death, despite their naturally high iron-dependence. The degree of ferroptosis resistance varies within different cancer types, with some forms of cancer being highly susceptible to ferroptosis, while others appear highly resistant [9]. Exploiting the iron-rich nature of certain tumors by inhibiting ferroptosis resistance pathways offers a strategy to induce cell death in otherwise drug-resistant cancers.

To better understand mechanisms linked to ferroptosis sensitivity, we utilized TSC cell line models. TSC is a rare genetic disorder caused by the loss of function of either *TSC1* or *TSC2*. TSC patients are predisposed to kidney, skin, and brain tumors (see reviews [10,11]). A central feature of TSC and its associated cell line models is that mechanistic target of rapamycin (mTOR) signaling is hyperactive [12,13]. Recent findings underscore the mTOR pathway as a pivotal but context-dependent regulator of ferroptosis [13]. In some settings, oncogenic mTOR activity inhibits ferroptosis by promoting GPX4 translation and SREBP-mediated lipogenesis, thereby bolstering lipid membrane stability [14,15,16]. Conversely, under certain conditions, mTOR can facilitate ferroptosis via autophagic and lysosomal mechanisms that elevate intracellular iron or potentiate LPO [17]. These contrasting outcomes reflect the complexity of mTOR-driven metabolic and redox pathways, suggesting that the net effect of mTOR activation on ferroptosis susceptibility is dictated by cell type, oncogenic drivers, and the broader metabolic environment. In this study, we explored ferroptosis survival mechanisms in both TSC and cancer cell lines to better understand potential vulnerabilities that could be therapeutically targeted.

## 2. Materials and Methods

### 2.1. Cell Culture, Drugs and Reagents

TSC cell lines used in this study include *Tsc2*+/+ *p53*−/− and *Tsc2*−/− *p53*−/− mouse embryonic fibroblasts (MEFs) (now referred to as *Tsc2*+/+ and *Tsc2*−/−, respectively), which were kindly provided by D. Kwiatkowski in 2004 (Harvard University, Boston, MA, USA) [18]. Eker rat leiomyoma-derived cells lacking functional *Tsc2* (ELT3-V3) and matched controls that re-express *Tsc2* (ELT3-T3) were generated by Astrinidis et al. [19]; gifted in 2006 by C. Walker (M.D. Anderson Cancer Center, Houston, DX, USA). For ELT3 cells, detailed oncogenic mutation data are not available beyond the reported *Tsc2* inactivation. Human *TSC2*-deficient angiomyolipoma (AML) 621-101 cells (*TSC2*(−) AML) and these cells restored with re-expressed TSC2 (621-103 *TSC2*(+) AML) were a gift from E. Henske in 2006 (Harvard University, Boston, MA, USA) [20]. *TSC2*-deficient AML cells were derived from a patient with TSC and harbor a loss-of-function mutation in *TSC2*. While their broader genetic landscape has not been comprehensively profiled, previous studies on TSC patient-derived tumor cells indicate that these tumors are generally genetically stable, with *TSC1* or *TSC2* representing the predominant recurrent alterations [21]. In both AML and ELT3 models, re-expression of wild-type TSC2 provides an isogenic control framework, enabling phenotypic comparisons that can be attributed primarily to TSC2 status. OVCAR3, PEO1, PEO4 and MDA-MB-436 cells were purchased from American Type Culture Collection. PEO1 and PEO4 are derived from the same patient. PEO1 is characterized by a homozygous *BRCA2* mutation (c.5193C>G). *BRAC2* mutations give rise to hyperactivity of AKT/TSC2/mTORC1 signaling [22]. PEO4 has a secondary mutation (Y1655Y) that restores BRCA2 function. OVCAR3 is characterized by a mutation of *TP53*, and has increased *PIK3CA* transcription and reduced *PTEN* expression, which is known to enhance mTORC1 activation [23]. MDA-MB-436 contains *TP53*, *KRAS* and *PTEN* mutations. For instance, the MDA-MB-436 have a known *PTEN* p.V85_splice mutation resulting in mTORC1 hyperactivity [24].

Cells were grown in humidified incubators at 37 °C, 5% (*v*/*v*) CO_2_. Maintenance and treatments were carried out with 10% (*v*/*v*) fetal bovine serum (FBS, bought from Sigma-Aldrich, Dorset, UK) in Dulbecco’s modified Eagle’s medium (DMEM). Maintenance of *TSC2*(−) and *TSC2*(+) AML cells was performed with 15% (*v*/*v*) FBS. OVCAR3 cells were cultured in RPMI 1640 medium supplemented with 20% (*v*/*v*) FBS, 2 mM L-Glutamine, and 0.01 mg/mL insulin (Lonza, BE02-033E). PEO1 and PEO4 cells were cultured in RPMI 1640 medium supplemented with 15% (*v*/*v*) FBS and 2 mM L-Glutamine. MDA-MB-436 cells were cultured in DMEM supplemented with 15% (*v*/*v*) FBS, 0.01 mg/mL insulin, and 2 mM L-Glutamine. All culture medium was supplemented with 100 U/mL penicillin and 100 µg/mL streptomycin (P4333, Sigma-Aldrich). Unless otherwise stated, drugs to activate ferroptosis and inhibitors used in this study were bought from Selleck Biotechnology Ltd. (Waltham Abbey, UK). All other lab chemicals were purchased from Merck Life Science UK Limited (Watford, UK). Cells were routinely checked and were found to be mycoplasma negative with Venor™ GeM Advance Mycoplasma Detection Kits (Minerva Biolabs, Berlin, Germany).

### 2.2. Flow Cytometry

To quantify cell death by flow cytometry, cells cultured on plastic tissue plates were washed with phosphate-buffered saline (PBS) and then trypsinized to dissociate cells. Cells were collected and centrifuged at 2000 rpm (400× *g*) for 5 min. Cell pellets were then resuspended and incubated with 3 μM DRAQ7 for 10 min at 37 °C. Flow cytometry was performed using a BD Accuri C6. Excitation was performed with 633 nm, and emission was recorded with the FL4 filter [λem 675/25 nm]. DRAQ7 was sourced from Biostatus (Shepshed, UK).

To analyze caspase 3/7 activation and cell death, cells were seeded at 1 × 10^4^/100 μL/well into 96-well plates and assessed for confluence and attachment after 24 h in culture. RSL3 at 5 μM was applied to cells to induce ferroptosis and concomitantly treated with either no inhibitor, vehicle control, Deferiprone (100 μM), Ferrostatin 1 (10 μM), z-VAD-FMK (10 μM), or Necrostatin 1 (10 μM). After 24 h incubation at 37 °C, 5% CO_2_, humidified air, cells were assessed for caspase3/7 activity. Controls included untreated cells and untreated cells not exposed to the Caspase 3/7 probe. Supernatants were collected and monolayers removed by trypsinization, and both were pooled and centrifuged at 300× *g* for 5 min. CellEvent™ Caspase-3/7 (2 μM, bought from Thermo Fisher Scientific, Newport, UK) was prepared in fresh complete media, and cell pellets were resuspended in this solution before 30 min incubation at 37 °C, 5% CO_2_, humidified air. Analysis of probed cells was performed on a BD Accuri C6 [λex 488 nm, λem 533/30]. Gating was performed on FSC:SSC to exclude debris and include all cells for cell death analysis. Viability was defined as cells exhibiting a morphologically large size (FSC) and Caspase 3/7 negativity.

LPO was assessed with the Lipid Peroxide Detection Liperfluo (Dojindo Europe, Munich, Germany). After drug treatment, cells were trypsinized, collected, and pelleted. Cell pellets were then resuspended in culture medium containing Liperfluo (5 µM). Resuspended cells were then incubated for 30 min at 37 °C, 5% (*v*/*v*) CO_2_, before being stored on ice and assayed soon after. Flow cytometry was performed using a FACSCalibur^TM^ Flow Cytometer (BD, Cowley, UK) [λex 488 nm and λem 530/30 nm]. Readings were performed on a linear scale, and 15,000 gated events were recorded per sample. Gating was performed to exclude cellular debris and dead cells. Median fluorescent intensities were recorded. Data was processed using Floreada.io.

Metabolic markers to assess mitochondria and lysosomes and labile iron pool (LIP) were carried out using flow cytometry, as described above when analyzing caspase activation with various probes. JC1 probe was used for mitochondria membrane potential (2 mM [λex: 525 nm and λem: 590 nm]), MitoTracker™ to determine mitochondrial abundance (20 nM [λex: 488 nM and λem: 516 nm]), LysoTracker^®^ Red to assess lysosomal abundance (50 nM [λex: 577 nM and λem: 590 nm] was purchased from Thermo Fisher Scientific, and LysoSensor™ Green to measure lysosomal pH (1 mM [λex: 443 nM and λem: 505 nm]. BioTracker Far Red Labile Fe^2+^ (1 mM [λex: 646 nM and λem: 662 nm]) was purchased from Merck Millipore (Dorset, UK). When quantifying, background levels of unstained cells were subtracted from stained cells (*n* = 4).

### 2.3. siRNA Transfections

siRNA smartpools were purchased from Horizon Discovery Ltd. and reconstituted to 100 µM in siRNA buffer (Horizon Discovery Ltd., Cambridge, UK). For cell viability assays, cells were seeded at 10,000 cells per well (96-well plates) and reverse transfection was carried out according to the manufacturer’s instructions. The final working concentration of siRNA for cell viability assays was 10 nM, and 0.15 µL of Dharmafect-1 transfection reagent was used per well. Cells were subject to knockdown for 48 h (or until 70–80% confluent) prior to other treatments. Target knockdown was verified via western blot and qPCR after 48 and 24 h, respectively.

### 2.4. Cell Viability Assays Using Crystal Violet

To assess cell viability by relative cell quantity determination, cells were grown in clear-bottom 96-well plates. A total of 4 technical repeats (i.e., 4 separate wells) were used per condition, per biological repeat. After treatment, wells were washed once with PBS. Next, wells were stained with crystal violet solution (0.5% (*w*/*v*) crystal violet, 20% (*v*/*v*) methanol in ddH_2_O) and incubated for 10 min on an orbital shaker. Wells were then washed 3 times with PBS, prior to resuspension in 1% (*w*/*v*) SDS solution. Plates were then incubated for 2 h on an orbital shaker before absorbance was measured at 570 nm on a PHERAstar^®^ FS plate reader (BMG LabTech, Buckinghamshire, UK). Blanks were subtracted, and data were expressed relative to untreated DMSO control cells that were assigned a value of 100%.

### 2.5. RNA Sequencing and Analysis

Four biological repeats were analyzed per condition. Cells were subject to reverse transfection with siRNA SMARTpools (DharmaconTM ON-TARGETplus siRNA) targeting *NFE2L2*, *AIFM2*, or non-targeting siRNA (all at 50 nM), using DharmaFECT-1 transfection reagent (DharmaconTM, Horizon Discovery Ltd.). After 72 h target knockdown, samples were collected in RNAprotect reagent (Qiagen, West Sussex, UK) prior to total RNA extraction with an RNEasy kit (Qiagen). Total RNA was quantified using a QubitTM RNA HS Assay. RNA samples for MDA-MB-436 cells were shipped to Novogene for RNA-sequencing and bioinformatic processing.

In brief, samples passed through quality control and strand-specific and non-strand-specific libraries were prepared and validated for quality control. Libraries were pooled and subjected to Illumina sequencing. Reads were quality trimmed with FastP [25] (the exact software version was not specified by Novogene) before mapping with HISAT2 (2.2.1) [26]. Counts were then assigned with FeatureCounts (2.0.6) [27]. The DESeq2 R package (1.42.0) was used to perform differential expression analysis between groups [28]. Comparisons were made for all groups versus the non-targeting siRNA-treated group. *p*-values were corrected for multiple testing using the Benjamini-Hochberg false discovery rate method.

Gene Ontology (GO) analysis was performed in-house using GeneAnalytics (https://geneanalytics.genecards.org/). GO for ferroptosis gene sets was collated from the literature searches. The NRF2-target gene list (sourced from [29]) was supplemented with additional targets based on literature searches. RNA sequencing data used for volcano plots comparing 621-101 *TSC2*(−) AML versus TSC2 re-expressed 621-103 *TSC2*(+) AML cells, and the ferroptosis and NRF2-target genes sets are provided in Appendix A. RNA samples preparation and RNA sequencing for TSC-model cell lines (supported through Wales Gene Park (Cardiff University)) is described in [30]. We reanalyzed RNA sequencing data for ferroptosis and NRF2-target genes comparing *Tsc2*−/− and *Tsc2*+/+ MEFs (found in Appendix A of previously published manuscript [30]).

Raw RNA sequencing data comparing MDA-MB-436 with NRF2 knockdown versus MDA-MB-436 cells treated with non-targeting siRNA are provided in Appendix A. Appendix A shows raw RNA sequencing data comparing MDA-MB-436 with FSP1 knockdown versus MDA-MB-436 cells treated with non-targeting siRNA.

### 2.6. Western Blotting

Cell lysates were prepared by direct cell lysis cells using x1 sample buffer (62.5 mM Tris–HCl (pH 7.6), 50 mM dithiothreitol, 2% (*w*/*v*) sodium dodecyl sulfate, 10% (*v*/*v*) glycerol, and 0.1% (*w*/*v*) bromophenol blue), sonicated for cycles of 3 × 30 s at full power (30 sonication amplitude microns (µm)), heated for 10 min at 95 °C, and centrifuged for 8 min at 13,000 rpm (16,000× *g*). To quantify protein, Pierce 660 nm protein reagent (supplemented with ionic detergent compatibility reagent) was used as directed by the manufacturer (Thermo Fisher Scientific, Newport, UK). An equal protein amount was loaded on NuPage precast gels (Thermo Fisher Scientific, Newport, UK). Western blotting was performed as previously described [31]. Blot images were processed on ImageJ (v.50) with minimal alterations to contrast and brightness levels only. Supplementary Uncropped Blots are provided in the Appendix A. Antibodies were purchased from Cell Signaling Technology (Danvers, MA, USA), including antibodies towards TSC2 (#4308), TfR1 (#55487), GPX4 (#52455) and β-actin (#4967). Antibodies for HMOX1 were purchased from Abcam Limited, Cambridge, UK (#ab68477) and for FSP1 from Proteintech Europe, Manchester, UK (#20886-1-AP).

### 2.7. Immunofluorescence

Cells were grown on glass coverslips. After treatment, cells were washed in PBS for 5 min and then fixed with 4% (*v*/*v*) paraformaldehyde in PBS for 15 min (all wash/incubation steps were at room temperature, unless stated otherwise). Cells were washed 3 times with PBS (5 min incubations) before being quenched with 0.1 M glycine in PBS for 15 min. After quenching, cells were washed in PBS and then permeabilized with 0.2% (*v*/*v*) Triton X–100 in PBS for 20 min. Following permeabilization, the buffer was removed, and cells were blocked with 2% (*w*/*v*) bovine serum albumin in PBS for 20 min. Primary NRF2 antibody (#PA5-27882 (Thermo Fisher Scientific)), diluted 1:500 in blocking buffer, was then added to the cells and incubated overnight at 4 °C. Following three 5 min washes in PBS, the secondary antibody (polyclonal donkey anti-rabbit Alexa Fluor 647, #A-31573 (Thermo Fisher Scientific)) was diluted 1:1000 in blocking buffer and applied to the cells for 30 min. Cells were then washed 3 times in PBS for 5 min before incubating with DAPI (1 μg/mL in PBS) for 15 min. Cells were washed twice for 5 min with PBS. Coverslips were mounted with 5 μL Fluoromount G mountant. Cells were examined on a Leica DMI6000B microscope (Leica Biosystems, Wetzlar, Germany; DAPI [λex 340–380 nm and λem 450–490 nm]; Far Red [λex 590–650 nm and λem 662–738 nm]) with fixed exposure time for comparative fluorescence intensity.

### 2.8. Statistical Analysis

The number of biological and technical replicates was tailored to the specific assay type, balancing scientific rigor, reproducibility, and resource considerations. For western blot experiments, three biological replicates were used, which is standard for protein quantification where signal-to-noise is high and reproducibility across experiments is consistent. RNA-seq experiments were performed with 4–6 biological replicates to ensure adequate statistical power for differential gene expression analysis and to align with current best practices in transcriptomics. Cell viability assays were conducted with three biological replicates, each containing multiple technical replicates, to account for pipetting variability and to identify potential artefacts (e.g., air bubbles) inherent to high-throughput plate-based formats. All replicate numbers are stated in the figure legends and were selected to provide sufficient statistical power for the analyses performed. Experimental work was conducted in batch format, with biological replicates grown in parallel within each batch (cell passage number maintained below 30). To ensure reproducibility, selected assays were repeated in independent experimental batches. While these confirmatory batches supported the overall trends, they did not always include the full set of controls or drug treatment conditions used in the original experiments. GraphPad Prism 9 was used for statistical analysis. If the data had a normal (Gaussian) distribution, one-way ANOVA with Tukey’s post hoc test was used. If examining the effects of two independent variables, two-way ANOVA with Šídák’s multiple comparisons test was used. *p*-values are represented as either * *p* < 0.05, ** *p* < 0.01, *** *p* < 0.001, **** *p* < 0.0001 or as being not significant ‘NS’. Data are presented as the mean ± SEM.

## 3. Results

### 3.1. TSC2-Deficiency Protects Cells from Ferroptosis Inducers

We examined the effects of ferroptosis inducers on cell line models of TSC. The ferroptosis inducers, RSL3 and erastin, were used in 24 h treatments with *Tsc2*(−/−) and *Tsc2*(+/+) MEF cells (Figure 1a) and *Tsc2*-null ELT3-V3 and TSC2 re-expressed ELT3-T3 cells (Figure 1b). Cell viability was assessed by flow cytometry using DRAQ7 labelling that measures cell death via increased membrane permeability.

RSL3 induces ferroptosis through direct inhibition of GPX4 without GSH depletion, and more recently has been shown to inhibit the catalytic activity of thioredoxin reductase 1, which further contributes to LPO and ferroptosis [32]. Erastin induces ferroptosis via inhibition of the cysteine-glutamate transporter Xc(−) system that prevents cysteine import for GSH regeneration, which then reduces the cell’s capacity to reduce lipid peroxides [33]. When compared to their wild-type controls, *Tsc2*-deficient cells were less sensitive to ferroptosis when treated with either of the ferroptosis inducers, RSL3 or erastin (Figure 1a,b). Treatment with RSL3 (0.3 μM) and erastin (1.2 μM) reduced viability in the *Tsc2*(−/−) MEF cells by ~50%, while showing ~75% loss of cell viability in the *Tsc2*(+/+) MEF cells. Similarly, higher concentrations of ferroptosis inducers were required to kill the *TSC2*-deficient ELT3-V3 cells.

As a third TSC model, *TSC2*(+) and *TSC2*(−) AML were treated with RSL3 and cell viability was assessed (Figure 1c). Ferrostatin-1 was employed to inhibit ferroptosis as a control. The *TSC2*(−) AML cells showed resistance to ferroptosis-induced cell death, when compared to the *TSC2*(+) control cells. Ferrostatin-1 (a lipid ROS scavenger) rescued cell death, showing that RSL3 is inducing cell death via ferroptosis. To further confirm that treatment was inducing ferroptosis in an iron-dependent way, a panel of inhibitors was used in combination with RSL3 in the *Tsc2*(+/+) MEF, and cell viability was determined (Figure 1d). To explore another TSC model, wild-type cell line, *TCS2*(+) AML cells were analyzed in cell viability assays (Figure 1e). Inhibitors of different mechanisms of cell death used included: (i) the iron chelator, deferiprone (DFP), (ii) ferrostatin-1, to inhibit ferroptosis, (iii) pan-caspase inhibitor, Z-VAD-FMK, to inhibit ‘classic’ apoptosis, and (iv) necrostatin-1, to inhibit necroptosis. Deferiprone, ferrostatin-1 and necrostatin-1 ablated RSL3-induced ferroptosis, while Z-VAD-FMK was unable to rescue RSL3-induced ferroptosis. Phase contrast images of the *TSC2*(+) AML cells seeded at low cell density were taken to illustrate that RSL3 treatment caused the cells to round up (Figure 1f), which was prevented with either deferiprone, ferrostatin-1 or necrostatin-1. In the *Tsc2*(+/+) MEF and ELT3-T3 (*Tsc2*+) cells, we show that RSL3-induced cell death occurs in the absence of caspase-3/7 activation, which is consistent with ferroptosis (Appendix A, respectively). Flow cytometry showed a loss of viable cells based on FSC/SSC gating, while a caspase-3/7 activity probe showed only a weak increase in fluorescence that did not represent caspase-3/7 activation (Appendix A). Cell death was fully rescued by ferrostatin-1 and DFP, but not by Z-VAD-FMK, confirming that death was caspase-independent and iron-dependent. Of interest, necrostatin-1 rescued RSL3-induced death in both the MEF and AML cells, which could be due to necrostatin-1 having radical scavenging activity, which has been reported to rescue ferroptosis indirectly [34]. Collectively, these experiments demonstrate that loss of viability caused by RSL3 is iron-dependent, i.e., by ferroptosis.

To further explore ferroptosis induction in these TSC cell line models, we carried out differentially expressed gene (DEG) analyses of ferroptosis-linked genes from transcriptomic data. We compared differences in expression of ferroptosis genes between *Tsc2*(−/−) and *Tsc2*(+/+) MEFs (Figure 2a) and *TSC2*-deficient and *TSC2*-restored AML cells (Figure 2b). In cells lacking *TSC2*, we observed an upregulation of ferroptosis gene expression. Of note, *HMOX1* expression was upregulated in the absence of *TSC2*. We also observed upregulation of the Xc(−) system cystine/glutamate transporter, *SLC7A11*, which is directly linked to ferroptosis resistance and a drug target of erastin to block SLC7A11-mediated cystine import [35]. We next examined the relative protein expression of targets linked to ferroptosis in these TSC cell models: *Tsc2*(+/+) and *Tsc2*(−/−) MEF (Figure 2c), ELT3-T3 (*Tsc2*+) and ELT3-V3 (*Tsc2*−) (Figure 2d) and *TSC2*(+) and *TSC2*(−) AML cells (Figure 2e). Ferroptosis-related proteins examined were HMOX1, GPX4, GPX8, transferrin receptor (TfR1, gene referred to as *TFRC*) and FSP1. Although RSL3 treatment did not alter expression of ferroptosis-related proteins, *TSC2*-deficiency consistently increased levels of HMOX1. Densitometry analysis revealed significant upregulation of HMOX1 protein expression by 16-fold (*p* = 0.0148), 9-fold (*p* < 0.0001), and 36-fold (*p* = 0.0032) in *TSC2*-deficient MEF, ELT3, and AML cells, respectively, compared with their wild-type counterparts after RSL3 treatment (Appendix A). *Tsc2*-deficient MEF cells also exhibited significant increases in TfR1 and FSP1 protein levels relative to their wild-type controls. In addition, GPX4 protein expression was elevated by 1.7-fold (*p* = 0.0299) in *TSC2*-deficient AML cells following RSL3 treatment. Collectively, these findings indicate that loss of *TSC2* induces upregulation of multiple ferroptosis-associated proteins, which may contribute to the observed resistance to ferroptosis.

We next used flow cytometry to assess the LIP and functional markers of mitochondria and lysosomes, as these metabolic parameters are closely linked to ferroptosis (Appendix A). LIP was elevated in *TSC2*-deficient AML and ELT3 cells but was reduced in the *Tsc2*−/− MEFs. All three *TSC2*-deficient models exhibited higher mitochondrial load, consistent with mTORC1-driven mitochondrial biogenesis via YY1 transcription factor [36] and impaired mitophagy [37]. Elevated mitochondrial load was associated with increased oxidative phosphorylation, as evidenced by JC-1 staining of mitochondrial membrane potential (ΔΨm) [38]. Lysosomal load was increased in the AML and ELT3 cells lacking *TSC2*, while lysosomal acidity was elevated in all *TSC2*-deficient lines, although this did not reach statistical significance in the ELT3 cells.

These alterations have potentially opposed effects on ferroptosis susceptibility. Increased mitochondrial mass and oxidative phosphorylation are expected to elevate ROS, which can promote lipid peroxidation [39]. In the AML and ELT3 cells, the combination of high LIP and increased lysosomal acidity suggests enhanced ferritinophagy and iron release, potentially sensitizing cells to ferroptosis. In contrast, the *Tsc2*−/− MEFs exhibited lower LIP despite high mitochondrial activity, suggesting preferential iron sequestration into iron–sulfur clusters, which may confer relative resistance unless iron availability is increased or GPX4 is inhibited. Together, these findings suggest that *TSC2* loss can prime mitochondria for ROS generation and alter iron handling, with the ferroptotic outcome determined by the balance between iron sequestration and lysosomal iron release.

### 3.2. NRF2 Protects TSC2-Deficient Cells from Ferroptosis

Previous studies suggest that loss of *TSC2* elevates oxidative stress [40,41]. This chronic oxidative environment may trigger compensatory upregulation of ferroptosis defense pathways, e.g., enzymes involved in lipid peroxide detoxification, thereby reducing sensitivity to ferroptosis. As homeostatic balance of oxidative stress is possibly modulated by the transcription factor NRF2 during induction of ferroptosis, we carried out DEG analyses of NRF2 target genes from *Tsc2*(−/−) compared to *Tsc2*(+/+) MEFs (Figure 3a) and *TSC2*-deficient and *TSC2*-restored AML cells (Figure 3b).

In both TSC cell line models, we observed an upregulation in the expression of NRF2 target genes upon *TSC2* loss, highlighting that NRF2 is likely activated in these cells. In both cells, NRF2-target genes *HMOX1*, *SLC7A11*, *SLC40A1*, *AIFM2*, *FTH1*, *FTL* and *SQSTM1* were upregulated. To examine NRF2 in the ELT3 cell line model of TSC, we examined NRF2 localization in cells by confocal microscopy (Figure 3c). DAPI was employed as the nuclear counterstain. Secondary antibody control images, showing no background non-specific staining, are shown in Appendix A. NRF2 was predominantly localized to the cytoplasm in untreated conditions, while RSL3 treatment resulted in nuclear translocation of NRF2. Nuclear translocation of NRF2 was much more pronounced in *Tsc2*-deficient ELT3 cells (ELT3-V3), when compared to the wild-type *Tsc2*(+) controls (ELT3-T3). A higher level of nuclear staining of NRF2 upon RSL3 treatment in *Tsc2*-deficient ELT3 cells (ELT3-V3) is indicative of NRF2 activation. Densitometry analysis indicated that 27.6 ± 6.6% of NRF2 was nuclear in untreated conditions and that it rose to 55.3 ± 8.9% upon RSL3 treatment (*p* = 0.0135 *).

We reasoned that in these *TSC2*-deficient cells, NRF2 could play a role in their resistance to ferroptosis. To explore this possibility, we used an NRF2 inhibitor, ML385. Treatment with ML385 was compared in *Tsc2*(−/−) MEFs (Figure 3d) and *TSC2*(−) AML cells (Figure 3e) that were treated with RSL3 to induce ferroptosis. NRF2 inhibition was found to resensitize both *TSC2*-deficient cell lines to RSL3-induced ferroptosis, where loss of cell viability by ferroptosis occurred at lower doses of RSL3 when combined with ML385. This data shows that NRF2 protects *TSC2*-deficient cells from ferroptosis. As a control to verify that RSL3 was inducing iron-dependent ferroptosis, we used ferrostatin-1 in a related experiment to rescue both *Tsc2*(−/−) MEFs and *TSC2*(−) AML cells from ferroptosis (Appendix A).

We next considered whether a transcriptional gene target of NRF2, *AIFM2,* was involved. AIFM2 encodes FSP1, a key anti-ferroptotic enzyme that functions independently of GPX4 to prevent LPO. FSP1 localizes to the plasma membrane, where it reduces ubiquinone (CoQ10) to its antioxidant form, ubiquinol, thereby neutralizing LPOs and protecting cells from ferroptosis [42]. Previous studies suggest that NRF2 activation upregulates FSP1 expression as part of its antioxidant response program [43]. Given our findings that NRF2 is activated in *TSC2*-deficient cells, we next explored whether FSP1 contributes to ferroptosis resistance. We utilized a FSP1 inhibitor (iFSP1), where we observed that iFSP1 treatment resensitized *Tsc2*(−/−) MEFs to RSL3-induced ferroptosis (Appendix A). However, iFSP1 did not resensitise *TSC2*(−) AML cells to RSL3-induced ferroptosis (Appendix A), highlighting a difference in dependency of ferroptosis survival genes between TSC cell line models. This data indicates a role for NRF2 in promoting ferroptosis resistance in these cells, and that most likely involves mechanisms that are not downstream of FSP1.

### 3.3. FSP1 and NRF2 Mediate Ferroptosis Resistance in Cancer Through Independent Mechanisms

Our findings indicate that NRF2 and FSP1 can both contribute to ferroptosis resistance in *TSC2*-deficient cell line models. However, it remains unclear whether these targets linked to ferroptosis resistance are unique to TSC or whether this also extends to cancer. Many cancer types exhibit high levels of NRF2 expression or activity linked to redox homeostasis, which is considered to be pro-tumorigenic and is associated with ferroptosis resistance [44,45]. Similarly, FSP1 functions as a major ferroptosis suppressor in cancer [46]. Ferroptosis was recently suggested as a potential therapeutic target in both ovarian [47] and breast cancers [48]. Therefore, we next examined the dependency of FSP1 and NRF2 to resist ferroptosis in ovarian and breast cancer cell lines.

Instead of using iFSP1, which may have off-target effects, we carried out siRNA knockdown of FSP1. Validation of effective FSP1 siRNA knockdown when compared to a non-target siRNA control was determined by quantitative real-time PCR (RT-PCR). Appendix A shows that FSP1 siRNA was effective in knocking down FSP1 expression in HEK293 and a panel of ovarian cancer cells (PEO1, PEO4 and OVCAR3). In PEO1, PEO4, and OVCAR3 cells, a tolerable dose of RSL3 (30–50 nM) was used for each cell line. Cell viability assays were carried out to compare cells with and without FSP1 knockdown. FSP1 knockdown markedly sensitized these ovarian cell lines to RSL3-induced ferroptosis (Figure 4a–c). Consistent with enhanced ferroptosis sensitivity, phase contrast images show that the OVCAR3 cells round up and shrink with FSP1 knockdown and RSL3 treatment.

As a highly resistant cancer cell line to RSL3-induced ferroptosis, we examined MDA-MB-436 cells. These breast cancer cells tolerated a high 1 μM dose of RSL3, showing no loss of cell viability (Figure 4d). We observed that targeting NRF2 for knockdown increased the level of RSL3 sensitivity in MDA-MB-436 cells by a moderate degree (where we observed a loss of ~30% cell viability). To examine how NRF2 might protect these cells against ferroptosis in more detail, we compared RNA sequencing data from NFE2L2 siRNA versus non-target siRNA. NRF2 knockdown caused 24 upregulated genes and 34 downregulated genes that were > and <than 2-fold (Figure 4e). DEG analysis indicated several key genes linked to ferroptosis survival. This includes *SLC7A11*, which would diminish the reduced form of GSH to enhance sensitivity to ferroptosis. Other key targets that were downregulated upon NFR2 knockdown are annotated in the volcano plot and included *TXNRD1*, *PIR*, *HMOX1*, *FTH1* and *FTL*. As expected, *NFE2L2* was markedly reduced in its expression, showing that NRF2 was effectively knocked down in this experiment. GO terms that correlated to these downregulated genes corresponded most strongly to “Negative Regulation of Ferroptosis”, and these specifically included *NQO1*, *SLC7A11*, *HMOX1*, and *FTH1*. Also included were “Response to Oxidative Stress”, “Intracellular Sequestering of Iron Ion”, and “Leukotriene Transport”, the latter of which relates to metabolism of polyunsaturated fatty acids (PUFAs), which are an essential initiating component of ferroptosis [49]. The most strongly correlated GO pathway was “Ferroptosis”, which included the above-mentioned genes as well as *AKR1C1* and *GCLM*. Within the upregulated genes after NRF2 knockdown, there were three highly matched upregulated pathways that were focused on extracellular matrix organization.

Given that NRF2 knockdown only partially sensitized MDA-MB-436 to RSL3, we wanted to assess whether FSP1 knockdown might be more effective. Indeed, FSP1 knockdown caused loss of cell viability by >95% when treated with RSL3 (Figure 4f).

To visualize the differences of ferroptosis induction, we carried out time-lapse videos of the MDA-MB-436 cells that were treated with RSL3 over an 18 h period and with RSL3 treatment after FSP1 knockdown (Appendix A). This data indicates that FSP1 is critically involved in protecting these cells from ferroptosis, making it a promising therapeutic target for ferroptosis-resistant cancer cells. We compared RNA sequencing data from *AIFM2* versus non-target siRNA. *AIFM2* expression was markedly reduced, showing that the *AIFM2* siRNA was effective. FSP1 knockdown only resulted in a moderate change to their gene expression profile, causing a small number of genes to be changed in expression, i.e., 15 upregulated genes and 3 downregulated genes that had changed by −/+ 2-fold (Figure 4g). Using a lower fold-change threshold, we cross-compared NRF2-target genes affected by knockdown experiments. Genes downregulated by *NRF2* knockdown were modestly upregulated following *FSP1* knockdown, suggesting that loss of *FSP1* enhances NRF2-dependent gene expression (Figure 4h). This is likely a compensatory mechanism. Six NRF2-target genes were identified using a 0.5 Log2 Fold change analysis, and 15 with a lower stringency of 0.2 log2 fold change. Notably among them are the following NRF2-target genes: *NQO1*, *FTH1*, *TXNRD1*, *SLC7A11*, *ABCC2* and *ABCC3*, indicating that loss of FSP1 causes an increase in NRF2 transcriptional products. Within that set, *NQO1*, *TXNRD1* and *SLC7A11* are directly related to antioxidant response elements, suggesting FSP1 loss is compensated by upregulating other antioxidant systems. Links to increased iron sequestration can be made through *FTH1*, suggesting that cells are reducing sources of oxidative stress. *ABCC2* and *ABCC3*, which mediate ATP-dependent efflux of conjugated anions, have been implicated in leukotriene transport, linking them to the redistribution of PUFA-derived lipids, such as arachidonic acid, within cells, a process relevant for ferroptosis initiation [49]. Possibly, the higher increase in NRF2 transcriptional products could be through the upregulation of *PANX2* (Pannexin 2) that occurs after FSP1 knockdown. PANX2 has been linked to upregulation of NRF2 by an undetermined mechanism in prostate cancer [50].

To confirm that loss of cell viability was through enhanced LPO, we carried out LPO assays in both the OVCAR3 and MDA-MB-436 cell lines. These cell lines were examined as they show high sensitivity to RSL3 when combined with FSP1 knockdown. As expected, we observed a higher degree of LPO upon FSP1 knockdown with RSL3 treatment (Figure 4i and overlay graphs are provided in Appendix A), highlighting the critical protective role that FSP1 plays to prevent the LPO buildup.

## 4. Discussion

Ferroptosis offers a potential alternative therapy that could be used independently or in combination with current treatments. It is an emerging iron-dependent form of cell death intrinsically linked to cancer metabolism and tumorigenesis. Iron-dependent metabolism provides cancer cells with a proliferative and survival advantage within the tumor microenvironment. Cancer cells often become hard-wired into non-aerobic glycolysis, utilizing ferroptosis pathways that generate higher levels of oxidative stress. This metabolic shift increases cancer dependency on specific ferroptosis survival pathways, enabling them to thrive in a metabolically challenging environment.

Elevated iron metabolism, along with increased oxidative stress, could be a contributing factor to TSC pathology. The mTORC1 pathway stimulates lipid synthesis and incorporates PUFAs into cellular membranes [16], which serve as substrates for LPO. This should make *TSC2*-deficient cells more susceptible to the accumulation of LPO and ferroptosis. However, these cells appear to have adapted to higher levels of oxidative stress through elevated NRF2-mediated gene expression. mTORC1 is known to regulate the antioxidant defense system, which is crucial for counteracting oxidative stress-induced LPO. For example, mTORC1 promotes the release of NRF2 from its inhibitor KEAP1 (Kelch-like ECH-associated protein 1) by phosphorylating p62 (also known as sequestosome 1). This phosphorylation increases the affinity of p62 for KEAP1, leading to KEAP1 sequestration and subsequent NRF2 translocation into the nucleus [51]. NRF2 activation plays a crucial role in enhancing cellular resilience against oxidative damage and ferroptosis [52]. p62 and NRF2 have previously been implicated in TSC model systems. Immunohistochemical analysis of SEGA tissue by Malik et al. found increased expression of NRF2 target genes, *GCLC*, *GCLM* and *HMOX1* [53]. Additionally, mTORC1 has been reported to regulate iron metabolism by modulating the expression of iron transport proteins, TfR1 and ferroportin, further increasing intracellular iron levels [54].

In addition to the established links between mTORC1 hyperactivation and ferroptosis [15,16], iron metabolism and ferroptosis may also be connected to other pathological features of TSC, including aberrant activity of signal transducer and activator of transcription 3 (STAT3) and hypoxia-inducible factor 1α (HIF-1α). We recently demonstrated that both redox-sensitive transcription factors, STAT3 and HIF-1α, depend on redox factor-1 (REF-1, also known as APEX1) [55]. Inhibition of REF-1 was sufficient to suppress the pathologically elevated STAT3 and HIF-1α transcriptional activity in TSC model cell lines. During oxidative stress, REF-1 becomes activated and interacts with STAT3 and HIF-1α, reducing critical cysteine residues in their transcriptional activation domains to enhance DNA-binding and transcriptional function (reviewed in [56]).

Iron metabolism is regulated by multiple mechanisms and transcriptional programs, involving not only NRF2, but also STAT3 and HIF-1α. For instance, in colon and lung cancer cell lines, STAT3 is upregulated by iron-dependent activation of cyclin-dependent kinase 1 [57,58], and it promotes the expression of GPX4, a key regulator of ferroptosis resistance [59]. HIF-1α is likewise a central mediator of iron uptake and metabolism (reviewed in [60]). Thus, the increased ferroptosis resistance and altered iron handling observed in *TSC2*-deficient cells are likely influenced by the combined, aberrant activities of NRF2, STAT3, and HIF-1α under conditions of elevated oxidative stress [61].

Our research has demonstrated that *TSC2*-deficient cells and various cancer cells exhibit variability in their ferroptosis resistance mechanisms. Cancer cells exploit the advantages of iron-dependent metabolism to support their survival. Ferroptosis resistance appears to be primarily driven by three main mechanisms: NRF2, FSP1 and the GPX4/GSH system (reviewed in [4]). FSP1 plays a principal role in reversing LPO within plasma membranes. The primary function of FSP1 is to reduce CoQ10 to ubiquinol (CoQ10H2). Ubiquinol then scavenges phospholipid peroxyl radicals, suppressing the propagation of phospholipid peroxidation and ferroptosis [42]. We observed that knockdown of FSP1 changed the transcriptome profile of MDA-MB-436 cells by a moderate degree, implying that drug therapies to inhibit FSP1 might not dramatically alter gene expression. Minimal global transcriptional effect after FSP1 inhibition might be beneficial in treatment, i.e., it is an indicator that side effects and toxicity to non-cancerous cells might only be minimal. However, we did see a modest increase in the expression of several NRF2-target genes upon FSP1 knockdown, which is presumably a compensatory survival mechanism to protect these cells from rising levels of LPO. Our results suggest that baseline NRF2 pathway activity in *TSC2*-null AML cells may render FSP1 dispensable for ferroptosis resistance. FSP1 knockdown in these cells induces a compensatory upregulation of NRF2 targets, potentially through redox imbalance or CoQ10/NAD(P)H disruption, which may counteract ferroptosis sensitization. This highlights a mechanistically distinct reliance on ferroptosis-resistance pathways across different cancer lineages.

We found that drug inhibition of NRF2 with ML385 resensitized *TSC2*-deficient cells to ferroptosis. While we use low concentrations of ML385 that are routinely employed to inhibit NRF2 activity, we acknowledge that off-target effects unrelated to NRF2 could potentially influence ferroptosis induction. However, the lack of cytotoxicity observed with ML385 alone, together with the corroborating data from NRF2 knockdown in MDA-MB-436 cells, supports the specificity of our findings. NRF2 knockdown had a pronounced effect on gene expression in MDA-MB-436 breast cancer cells, notably reducing multiple anti-ferroptosis genes, including *SLC7A11*, *NQO1*, *TXNRD1*, *GCLM*, and *AKR1C1*. Among these, SLC7A11 imports cystine for GSH synthesis, which is critical for GPX4-mediated detoxification of lipid peroxides. Loss of *GCLM*, also involved in GSH synthesis, may further impair GPX4 activity and promote sensitivity to ferroptosis-inducing agents such as cisplatin [62].

Interestingly, *TXNRD1*, which was recently identified as an off-target of RSL3, was also reduced, suggesting that NRF2 knockdown may mimic aspects of RSL3 treatment. Downregulation of FTH1 may increase labile iron pools, further promoting ferroptosis.

NQO1, like FSP1, contributes to antioxidant defense by regenerating ubiquinone and vitamin E, which suppresses LPO. Since some tumors may depend preferentially on either NQO1 or FSP1 for protection, differential sensitivity to NRF2 or FSP1 inhibition could reflect these dependencies. NQO1 inhibition may therefore represent an alternative to NRF2 blockade, with a narrower transcriptional footprint.

This study focused on a limited panel of ovarian and breast cancer models, which are emerging as tractable targets for ferroptosis-based therapies [47,48]. However, tumor-type-specific ferroptosis resistance mechanisms may vary. Our findings in *TSC2*-deficient models, which exhibit mTORC1 hyperactivation, suggest that dysregulated iron metabolism and ferroptosis resistance are hallmarks of mTORC1-driven cancers. Given that mTORC1 is hyperactive in up to 70% of cancers, these results highlight a potential therapeutic window to resensitize tumors to ferroptosis via targeted inhibition of NRF2, FSP1, or related pathways.

It should be noted that this study employed the classical necrostatin-1 analogue, which has been widely used in the literature but has also been reported to possess off-target antioxidant activity that may influence ferroptosis readouts [34]. While our results, together with iron chelation and ferrostatin rescue experiments, support the conclusion that RSL3 induces an iron-dependent form of cell death, future studies could employ the more specific necrostatin-1 analogue, Nec-1s, to better distinguish between effects mediated by RIPK1 inhibition and those arising from antioxidant activity.

A recent study focusing on KEAP1-mutant non-small cell lung cancer (NSCLC) found that FSP1 expression was partially dependent on NRF2 [63], though NRF2 inhibition alone did not significantly reduce FSP1 protein levels. Our data indicated that FSP1 expression was independent of NRF2 activity, suggesting that alternative regulatory mechanisms predominate. This indicates that dysregulated FSP1 expression in TSC tumors arises through NRF2-independent pathways. These findings highlight the importance of stratifying tumors by cancer type and expression profiles of key ferroptosis regulators when designing targeted therapies aimed at overcoming ferroptosis resistance.

## 5. Conclusions

Our work reveals that NRF2 and FSP1 contribute to ferroptosis resistance in *TSC2*-deficient, ovarian, and breast cancer cells. We show that inhibition of these pathways enhances ferroptotic sensitivity in a context-dependent manner, highlighting the need for patient stratification when designing ferroptosis-based therapies. Beyond identifying resistance factors, the next step is to consider how ferroptosis can be effectively induced in clinical settings. Potential strategies include targeted delivery of ferroptosis inducers using iron metabolism pathways, such as exploiting transferrin receptor-mediated uptake or nanoparticle-based systems that accumulate in iron-rich tumors. In addition, some standard cancer treatments, including radiotherapy, tyrosine kinase inhibitors, and chemotherapeutics such as cisplatin and sorafenib, have been shown to trigger ferroptosis or sensitize cells to ferroptosis induction. These could be repurposed or combined with direct inhibitors of GPX4 or FSP1 to enhance therapeutic efficacy. Moving forward, large transcriptomic datasets coupled with machine learning could be used to define ferroptosis and iron metabolism gene signatures (e.g., *NRF2*, *FSP1*, *SLC7A11*, and *GSH* biosynthesis enzymes), allowing for the development of predictive biomarkers and bespoke ferroptosis-inducing therapies. Overall, an improved understanding of ferroptosis resistance will enable the development of new therapeutic strategies for difficult-to-treat cancers, particularly those that resist conventional therapies.

## Figures and Tables

**Figure 1 cancers-17-02714-f001:**
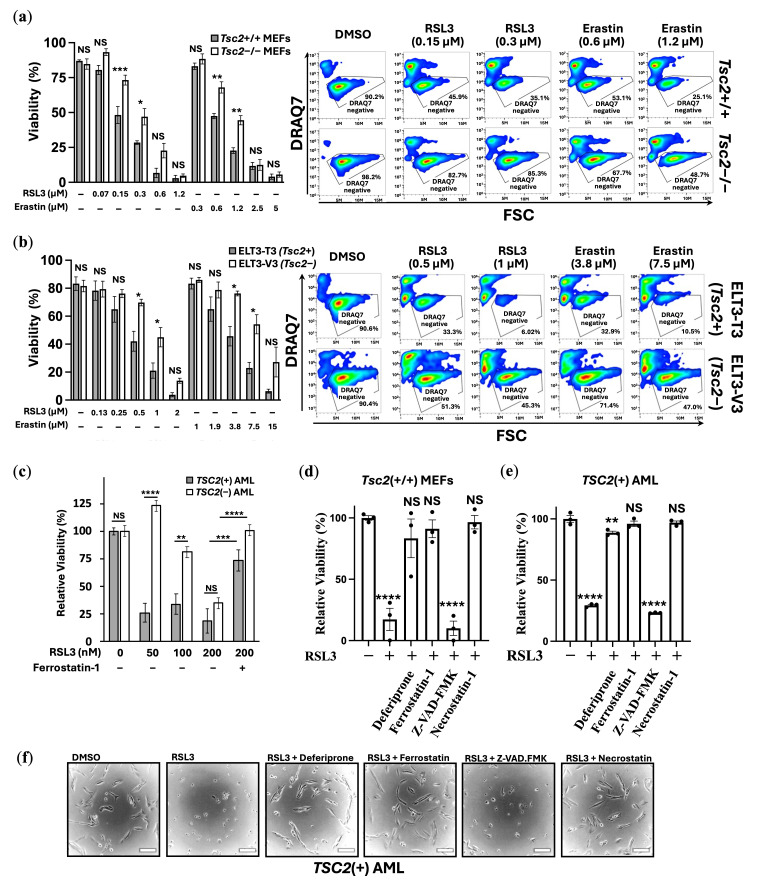
*TSC2*-deficient cells are resistant to ferroptosis induction. *TSC2*-deficient cell line models: (**a**) *Tsc2*+/+ versus *Tsc2*−/− MEF and (**b**) ELT3-T3 (*Tsc2*+) versus ELT3-V3 (*Tsc2*−), were subjected to flow cytometry following DRAQ7 staining after being treated with either DMSO vehicle, RSL3 or erastin at the concentrations indicated for 24 h. DRAQ7 exclusion (gated region) represents the viable cell population, while positive DRAQ7 staining (outside gated region) indicates cell death (representative flow images are shown). Cell viability (%) is graphed (*n* = 3). (**c**) Relative cell viability (%) of *TSC2*(+) and *TSC2*(−) AML cells were determined by cell counting with crystal violet stain after treatment with either DMSO or RSL3 (50, 100 and 200 nM) for 24 h (*n* = 3), and ferrostatin-1 (10 μM) was used as a control to prevent ferroptosis. Similarly, (**d**) *Tsc2*+/+ MEFs and (**e**) *TSC2*(+) AML cell viability was determined by cell counting with crystal violet stain. Cells were treated with either DMSO or RSL3 (100 or 200 nM, respectively) in the presence of either deferiprone (100 μM), ferrostatin-1 (10 μM), Z-VAD-FMK (10 μM) or necrostatin-1 (10 μM), as indicated, for 24 h (*n* = 3). (**f**) Phase contrast images of *TSC2*(+) AML cells at lower cell density were treated as above in panel E, to show morphological changes. Scale bar = 100 μm. (* = *p* < 0.05; ** = *p* < 0.005; *** = *p* < 0.0005; **** = *p* < 0.0001; NS = not-significant).

**Figure 2 cancers-17-02714-f002:**
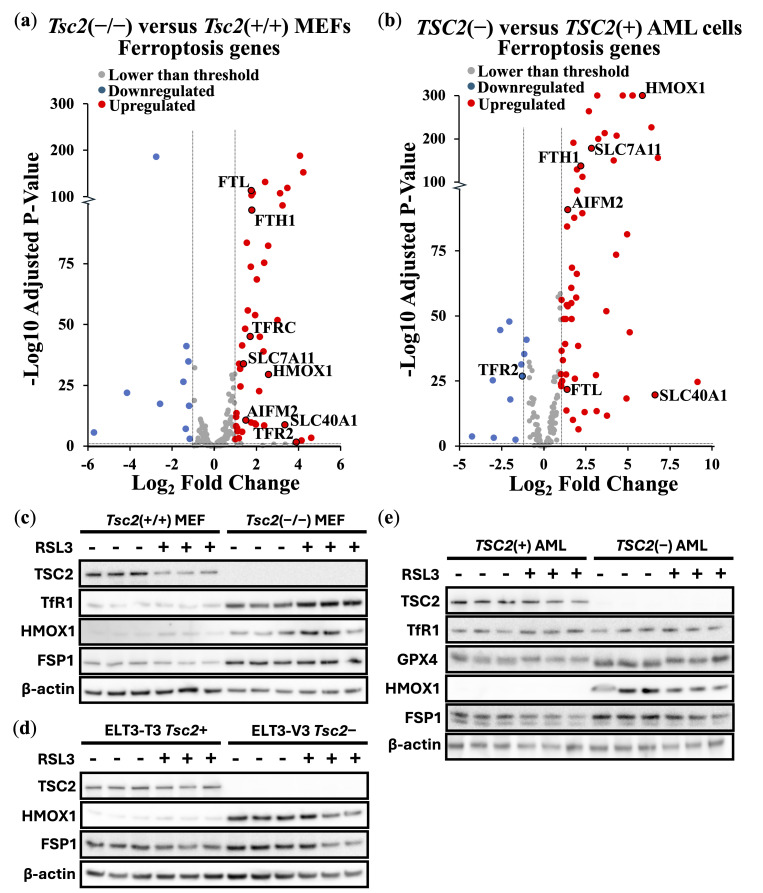
*TSC2*-deficient cells show enhanced expression of genes linked to ferroptosis. Transcriptomic data comparing (**a**) *Tsc2*−/− versus *Tsc2*+/+ MEF (*n* = 3) and (**b**) *TSC2*(−) versus *TSC2*(+) AML cells (*n* = 6), where ferroptosis-linked genes are shown in volcano plots. Thresholds were set at < and >2-fold changes (for down- and upregulated genes, respectively) with an adjusted *p*-value < 0.05. (**c**–**e**) Western blot analysis of ferroptosis associated proteins were also carried out on TfR1, GPX4, HMOX1 and FSP1 (depending on antibody specificity) in TSC cell line models (*Tsc2*+/+ and *Tsc2*−/− MEFs, ELT3-T3 (*Tsc2*+) and ELT3-V3 (*Tsc2*−), and *TSC2*(−) versus *TSC2*(+) AML cells, respectively) treated with either DMSO or 200 nM RSL3 for 4 h, as indicated (3 biological repeats of each shown). TSC2 and β-actin were analyzed as controls. The uncropped bolts are shown in Appendix A.

**Figure 3 cancers-17-02714-f003:**
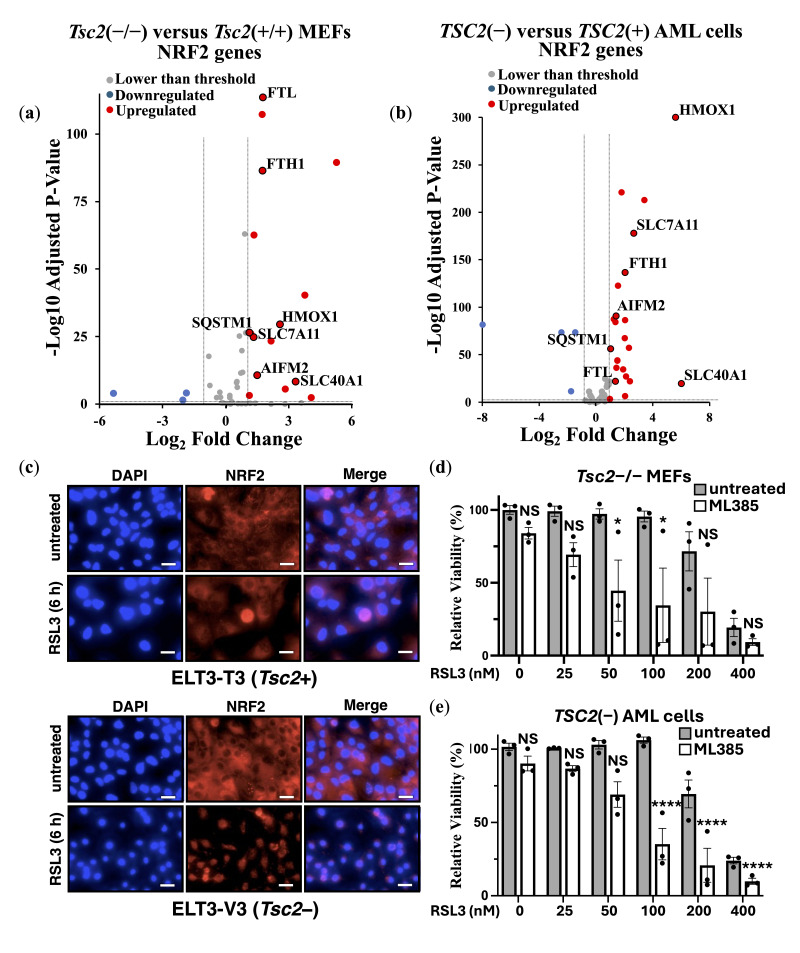
NRF2 is upregulated and protects *TSC2*-deficient cells from ferroptosis. Transcriptomic data comparing (**a**) *Tsc2−/−* versus *Tsc2*+/+ MEFs (*n* = 3) and (**b**) *TSC2*(*−*) versus *TSC2*(+) (*n* = 6), where NRF2-linked genes are shown in volcano plots. Thresholds of < and >2-fold changes and an adjusted *p*-value < 0.05 were set. (**c**) Confocal microscopy of ELT3-T3 (*Tsc2*+) and ELT3-V3 (*Tsc2−*) treated with either DMSO or RSL3 (6 h) was carried out using NRF2 antibodies (red) with nuclear counterstained with DAPI (blue). Both shown as ‘merge’, scale bar is 10 μm (*n* = 3). Relative viability (%) of (**d**) *Tsc2−/−* MEFs and (**e**) *TSC2(−)* AML cells was assessed by cell counting in crystal violet assays. Cells were treated with either DMSO or RSL3 (25–400 nM) in the presence or absence of ML385 (5 μM), as indicated, for 24 h (*n* = 3). (* = *p* < 0.05; **** = *p* < 0.0001; NS = not-significant).

**Figure 4 cancers-17-02714-f004:**
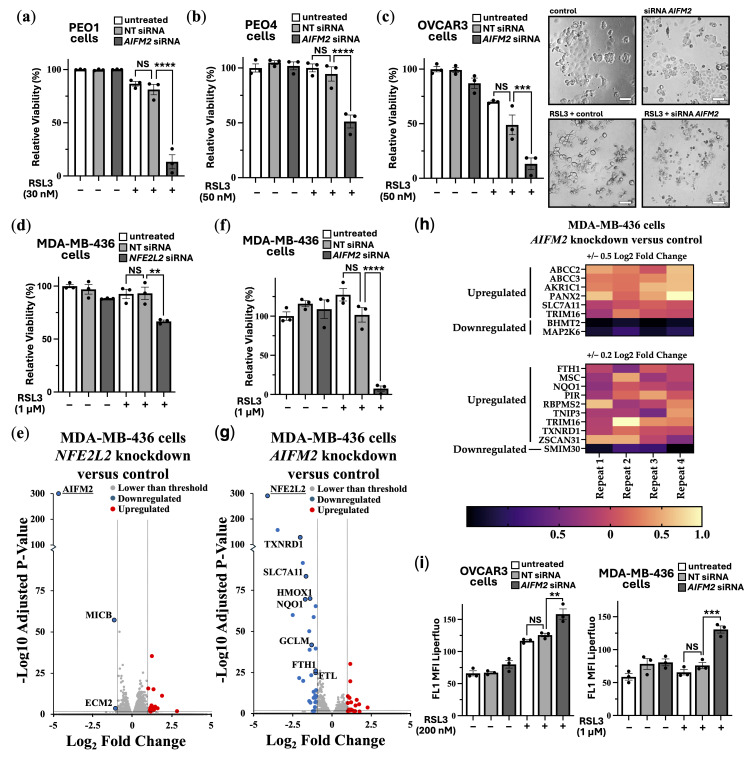
Ferroptosis resistance mechanisms in cancer cells involving NRF2 and FSP1. Relative viability (%) was determined by crystal violet assays in (**a**) PEO1, (**b**) PEO4 and (**c**) OVCAR3 cells with either non-target (NT) or *AIFM2* siRNA knockdown treated with either DMSO (control) or RSL3 at the indicated concentrations (*n* = 3). For OVCAR3 cells, phase contrast images at lower cell density are shown to depict morphological changes. Scale bar = 100 μm. (**d**) MDA-MB-436 cells were subjected to crystal violet assays to measure relative viability (%) after non-target (NT) or *NFE2L2* siRNA knockdown with or without RSL3 (1 μM) for 24 h (*n* = 3). (**e**) RNA sequencing data comparing MDA-MB-436 cells with *NFE2L2* siRNA knockdown versus non-target (*n* = 4) was carried out after 72 h. Thresholds were set (< and >2-fold changes and an adjusted *p*-value < 0.05). Respectively, (**f**,**g**) are identical to panels ‘d’ and ‘e’, apart from *AIFM2* siRNA knockdown rather than targeting NFE2L2. Heatmap comparing AIFM2 versus non-target siRNA knockdown on NRF2-linked genes in MDA-MB-436 cells is shown in (**h**). (**i**) LPO was assessed in OVCAR3 and MDA-MB-436 cells that were treated with either non-target (NT) or *AIFM2* siRNA knockdown with or without RSL3 (at either 200 nM or 1 μM) for 24 h (*n* = 3). (** = *p* < 0.005; *** = *p* < 0.0005; **** = *p* < 0.0001; NS = not-significant).

## Data Availability

All datasets generated or analyzed during this study are either included in this article or Appendix A. The data analyzed during the current study are available from the corresponding author upon reasonable request. Both MDA-MB-436 cells with either NRF2 or FSP1 knockdown can be found at Gene Expression Omnibus (GEO) under the accession number GSE300960 (approved 14 July 2025).

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
