# Peer review of "Targeting NRF2 and FSP1 to Overcome Ferroptosis Resistance in TSC2-Deficient and Cancer Cells"

_cancers, 2025, doi:10.3390/cancers17162714_

Round 1

Reviewer 1 Report (Previous Reviewer 1)

Comments and Suggestions for Authors

Manuscript is improved according to the comments as well and can be published in present form.

Author Response

We thank the reviewer for their helpful suggestions, which has helped to improve the study.

Reviewer 2 Report (Previous Reviewer 4)

Comments and Suggestions for Authors

The manuscript, "Targeting NRF2 and FSP1 to Overcome Ferroptosis Resistance in TSC2-Deficient and Cancer Cells," presents interesting findings regarding ferroptosis resistance mechanisms. However, several aspects require critical re-evaluation and further clarification to meet the standards of a rigorous academic journal.

  1. Figures 2c-e display Western blot data devoid of accompanying quantitative analysis (e.g., densitometry). The text's reference to "upregulation" or "increased levels" of specific proteins is subjective in the absence of objective quantification. Quantitative densitometric analysis of all Western blots shown in Figures 2c, 2d, and 2e, normalized to a loading control (e.g., β-actin), should be supplied. Statistical significance of the reported alterations in protein expression must be incorporated.
  2. The methods section (2.4, 2.5) specifies varying quantities of biological replicates for distinct investigations (e.g., "4 technical replicates," "3 biological replicates for each shown" in Western blots, "Four biological replicates" for RNA-seq, and "n=3" for viability assays). Although variances may be needed, a definitive rationale for these discrepancies or their potential impact on the overall robustness of the conclusions remains unclear.  Kindly elucidate the reasoning behind the differential allocation of biological replicates across various experimental tests.  What is the effect of these decisions on statistical power and result interpretation, particularly for assays with limited replicates?
  3. The research used multiple TSC2-deficient cell lines, including Tsc2-/- MEFs, ELT3-V3, and TSC2(-) AML cells. Although TSC2 expression is a defining feature, a thorough molecular characterization of these cell lines, beyond the indicated TSC2 status (such as p53 status noted for MEFs but not for others, or other pertinent oncogenic drivers), is not consistently detailed in the methods or results.  This is especially pertinent considering the intricate interactions of pathways such as mTOR.  In addition to TSC2 status, what other significant genetic or molecular attributes (e.g., p53 status for all cell lines, mTOR pathway activation levels, or other prevalent oncogenic alterations) have been validated for all TSC2-deficient and control cell lines included in this study?  Kindly furnish this information and examine its possible ramifications for the identified ferroptosis resistance mechanisms.
  4. The research use ML385 to suppress NRF2. Although this is a prevalent inhibitor, there is a paucity of debate or experimental validation regarding its selectivity and possible off-target consequences at the utilized concentrations, particularly in light of the intricate cellular milieu.  Has the specificity of ML385 for NRF2 been thoroughly verified in the cell lines and concentrations employed?  Are there any recognized off-target effects of ML385 that may complicate the interpretation of the results, and if so, how have these been mitigated or considered?
  5. The study indicates that FSP1 suppression "did not resensitize TSC2(-) AML cells to RSL3-induced ferroptosis" (lines 511-512), however FSP1 knockdown proved successful in ovarian and breast cancer cells (Figure 4a-c, f). This indicates a context-dependent function, however the fundamental molecular distinctions between different cell types remain inadequately clarified.  The observation that FSP1 knockdown enhances NRF2-dependent gene expression (lines 618-620) is described as a "compensatory mechanism," although further investigation of this interaction is constrained. What precise molecular pathways distinguish FSP1 reliance in TSC2(-) AML cells from that in ovarian and breast cancer cell lines?  Additionally, what are the specific molecular processes through which FSP1 knockdown results in a compensatory elevation of NRF2 transcriptional products, and how does this compensation affect overall susceptibility to ferroptosis?
  6. Considering the focus on iron metabolism, were direct assessments of intracellular iron concentrations or labile iron pools performed in all cell lines? Which specific iron transport proteins (e.g., ferroportin) or storage proteins (e.g., ferritin) were examined, and what were their expression levels and functional implications regarding ferroptosis resistance in each cell line?
  7. The author could use the following papers (If applicable)
    Identification of Novel Molecular Panel as Potential Biomarkers of PAN-Gastrointestinal Cancer
    Screening: Bioinformatics and Experimental Analysis
    The role of Sparasis Latifolia and Aerobic Exercise on Anti-inflammatory Activities and Regulation of Inflammatory Factors in the Intestine of Mice with Colon Cancer Under
    Sparassis latifolia and exercise training as complementary medicine mitigated the 5-fluorouracil potent side effects in mice with colorectal cancer: bioinformatics approaches
  1. Was a further experiment conducted to explicitly validate the radical scavenging action of necrostatin-1 at the concentrations employed in these cell lines? What were the outcomes?  What other studies could be conducted to distinguish between RIPK1 inhibition and radical scavenging as the principal mechanism underlying necrostatin-1's influence on RSL3-induced cell death in this context?

Author Response

Comment 1: Figures 2c-e display Western blot data devoid of accompanying quantitative analysis (e.g., densitometry). The text's reference to "upregulation" or "increased levels" of specific proteins is subjective in the absence of objective quantification. Quantitative densitometric analysis of all Western blots shown in Figures 2c, 2d, and 2e, normalized to a loading control (e.g., β-actin), should be supplied. Statistical significance of the reported alterations in protein expression must be incorporated.

Response 1: we agree that this is beneficial when describing the data. We have now provided this as Supplementary Figure 2, where we normalized to the loading control. We have also reported the statistical significance when describing the results in the text of the manuscript. ‘Densitometry analysis revealed significant upregulation of HMOX1 protein expression by 16-fold (p = 0.0148), 9-fold (p < 0.0001), and 36-fold (p = 0.0032) in TSC2-deficient MEF, ELT3, and AML cells, respectively, compared with their wild-type counterparts after RSL3 treatment (Supplementary Figure S2). Tsc2-deficient MEF cells also exhibited significant increases in TfR1 and FSP1 protein levels relative to their wild-type controls. In addition, GPX4 protein expression was elevated by 1.7-fold (p = 0.0299) in TSC2-deficient AML cells following RSL3 treatment. Collectively, these findings indicate that loss of TSC2 induces upregulation of multiple ferroptosis-associated proteins, which may contribute to the observed resistance to ferroptosis.’

Comment 2: The methods section (2.4, 2.5) specifies varying quantities of biological replicates for distinct investigations (e.g., "4 technical replicates," "3 biological replicates for each shown" in Western blots, "Four biological replicates" for RNA-seq, and "n=3" for viability assays). Although variances may be needed, a definitive rationale for these discrepancies or their potential impact on the overall robustness of the conclusions remains unclear.  Kindly elucidate the reasoning behind the differential allocation of biological replicates across various experimental tests.  What is the effect of these decisions on statistical power and result interpretation, particularly for assays with limited replicates?

Response 2: this is an important point raised to ensure scientific rigor and reproducibility. The number of replicates was tailored to the nature, technical variability, and resource intensity of each assay:

  • For western blots, we used 3 biological replicates, which is standard for protein quantification where the signal-to-noise ratio is relatively high and consistent results were observed across experiments.
  • RNA-seq was performed using four biological replicates to ensure adequate statistical power for differential expression analysis and to align with current best practices for transcriptomic reproducibility. Based on our experience and that of others, a range of 4-6 biological replicates are generally sufficient to detect meaningful gene expression changes linked to disease-relevant biology.
  • Cell viability assays were conducted with n=3 biological replicates, each with multiple technical replicates, which provides sufficient power given the low intra-assay variability and large effect sizes observed
  • The use of technical replicates in certain assays (e.g., four per condition in viability assays) was intended to ensure internal consistency and control for pipetting variability. This is particularly important in high-throughput, plate-based assays where such variability can influence results. This also helps mitigate artefacts such as those caused by air bubbles in individual wells, as these are more readily identified and discounted when multiple technical replicates are used. In contrast, assays such as western blotting are less prone to these specific technical errors and are typically interpreted at the level of biological replicates. Additionally, viability assays are relatively low-cost and performed in 96-well formats, making it practical and efficient to include multiple technical replicates within a single experiment to enhance reproducibility.

To enhance clarity, we have revised the Material and Methods and added: ‘Number of biological and technical replicates were tailored to the specific assay type, balancing scientific rigor, reproducibility, and resource considerations. For western blot experiments, three biological replicates were used, which is standard for protein quantification where signal-to-noise is high and reproducibility across experiments is consistent. RNA-seq experiments were performed with 4-6 biological replicates to ensure adequate statistical power for differential gene expression analysis and to align with current best practices in transcriptomics. Cell viability assays were conducted with three biological replicates, each containing multiple technical replicates, to account for pipetting variability and to identify potential artefacts (e.g., air bubbles), inherent to high-throughput plate-based formats. All replicate numbers are stated in the figure legends and were selected to provide sufficient statistical power for the analyses performed.’

Comment 3: The research used multiple TSC2-deficient cell lines, including Tsc2-/- MEFs, ELT3-V3, and TSC2(-) AML cells. Although TSC2 expression is a defining feature, a thorough molecular characterization of these cell lines, beyond the indicated TSC2 status (such as p53 status noted for MEFs but not for others, or other pertinent oncogenic drivers), is not consistently detailed in the methods or results.  This is especially pertinent considering the intricate interactions of pathways such as mTOR.  In addition to TSC2 status, what other significant genetic or molecular attributes (e.g., p53 status for all cell lines, mTOR pathway activation levels, or other prevalent oncogenic alterations) have been validated for all TSC2-deficient and control cell lines included in this study?  Kindly furnish this information and examine its possible ramifications for the identified ferroptosis resistance mechanisms.

Response 3: We agree that the molecular context of the cell models is important. The Tsc2-/- MEFs used in this study are also Tp53-null, which is a well-documented characteristic of this model. The TSC2(-) AML cells were derived from a patient with tuberous sclerosis complex. These cells harbor a loss-of-function mutation in TSC2, but their broader genetic landscape has not been comprehensively characterized. However, prior studies on TSC patient-derived tumor cells suggest that these cells derived from tumors are genetically stable, with TSC1 or TSC2 being the primary recurrent mutations [doi.org/10.1038/ncomms15816].

For ELT3 cells, their detailed oncogenic mutation profile has not been defined. Based on published sources, no other mutations in ELT3 have been specifically identified beyond the Tsc2 inactivation. Nevertheless, we believe that the re-expression of wild-type TSC2 in both AML and ELT3 cells provides a robust isogenic control framework. This approach allows us to attribute observed phenotypic differences, particularly ferroptosis sensitivity, to TSC2 status alone, thereby reducing confounding effects from unknown background mutations.

While we acknowledge that a full genetic characterization would add further depth, we believe the use of these established TSC2-null and TSC2-rescued pairs provides a solid and interpretable mechanistic basis for our conclusions.

This was added to the manuscript to the cell lines: ‘For ELT3 cells, detailed oncogenic mutation data are not available beyond the reported Tsc2 inactivation’ and ‘TSC2-deficient AML cells were derived from a patient with TSC and harbor a loss-of-function mutation in TSC2. While their broader genetic landscape has not been comprehensively profiled, previous studies on TSC patient-derived tumor cells indicate that these tumors are generally genetically stable, with TSC1 or TSC2 representing the predominant recurrent alterations [21 doi.org/10.1038/ncomms15816]. In both AML and ELT3 models, re-expression of wild-type TSC2 provides an isogenic control framework, enabling phenotypic comparisons that can be attributed primarily to TSC2 status.’

Comment 4: The research use ML385 to suppress NRF2. Although this is a prevalent inhibitor, there is a paucity of debate or experimental validation regarding its selectivity and possible off-target consequences at the utilized concentrations, particularly in light of the intricate cellular milieu.  Has the specificity of ML385 for NRF2 been thoroughly verified in the cell lines and concentrations employed?  Are there any recognized off-target effects of ML385 that may complicate the interpretation of the results, and if so, how have these been mitigated or considered?

Response 4: We appreciate the reviewer’s concern regarding the specificity of ML385. As with all small-molecule inhibitors, potential off-target effects must be considered. To minimize this, we used ML385 at concentrations commonly reported in the literature (between approximately 1 µM and up to 10 µM, with 5 µM being the most frequently reported and clearly effective dose (often observed to maximally inhibit NRF2 transcriptional activity in multiple cancer cell models)). ML382 has a IC50 of 1.9 μM. In our experiments, we employed doses at 5 μM. This is consistent with the current literature showing minimal toxicity even at 5 µM. In line with this, we observed that ML385 treatment alone did not reduce cell viability across our models, suggesting that at the concentration used, ML385 is not cytotoxic and unlikely to cause confounding effects due to general cellular stress or off-target toxicity. To further mitigate reliance on chemical inhibition alone, we also performed NRF2 knockdown experiments using siRNA in the cancer cell models. The consistency between genetic knockdown and pharmacological inhibition supports the specificity of our findings and strengthens the conclusion that NRF2 is a key regulator of ferroptosis resistance.

To address this point transparently, we have added the following statement to the Discussion section:

“We found that drug inhibition of NRF2 with ML385 resensitized TSC2-deficient cells to ferroptosis. While we use low concentrations of ML385 that are routinely employed to inhibit NRF2 activity, we acknowledge that off-target effects unrelated to NRF2 could potentially influence ferroptosis induction. However, the lack of cytotoxicity observed with ML385 alone, together with the corroborating data from NRF2 knockdown in MDA-MB-436 cells, supports the specificity of our findings.”

Comment 5: The study indicates that FSP1 suppression "did not resensitize TSC2(-) AML cells to RSL3-induced ferroptosis" (lines 511-512), however FSP1 knockdown proved successful in ovarian and breast cancer cells (Figure 4a-c, f). This indicates a context-dependent function, however the fundamental molecular distinctions between different cell types remain inadequately clarified.  The observation that FSP1 knockdown enhances NRF2-dependent gene expression (lines 618-620) is described as a "compensatory mechanism," although further investigation of this interaction is constrained. What precise molecular pathways distinguish FSP1 reliance in TSC2(-) AML cells from that in ovarian and breast cancer cell lines?  Additionally, what are the specific molecular processes through which FSP1 knockdown results in a compensatory elevation of NRF2 transcriptional products, and how does this compensation affect overall susceptibility to ferroptosis?

Response 5: We thank the reviewer for this insightful and multidimensional comment. We agree that our findings point to a context-dependent reliance on FSP1 for ferroptosis resistance, and that the compensatory increase in NRF2 transcriptional targets following FSP1 knockdown is a key mechanistic feature that warrants further study.

  1. Molecular distinctions between TSC2(-) AML and other cancer models:
    Our data suggest that TSC2(-) AML cells are
    less dependent on FSP1 compared to ovarian and breast cancer models, likely due to baseline activation of the NRF2 pathway in the TSC2-null setting. TSC2 loss is known to activate mTORC1, and our prior work and others' have shown that mTORC1 can promote NRF2 activity via multiple mechanisms, including inhibition of KEAP1-mediated degradation. As a result, TSC2(-) AML cells may rely more heavily on NRF2-driven glutathione and thioredoxin pathways for ferroptosis resistance, with FSP1 playing a more redundant role.
  2. FSP1 knockdown leads to compensatory NRF2 activation:
    We observed that FSP1 knockdown upregulates NRF2 target genes (e.g., GCLC, NQO1) specifically in TSC2(-) AML cells. While the precise signaling axis remains to be fully elucidated, this compensation could reflect a
    stress-induced activation of NRF2, possibly via mild lipid peroxidation, increased NAD(P)H demand, or altered redox buffering. Since FSP1 is an NAD(P)H-dependent oxidoreductase, its loss could shift redox balance or CoQ10 status, indirectly stabilizing NRF2 or enhancing its transcriptional activity. This compensation may blunt the ferroptosis-sensitizing effect of FSP1 knockdown in this specific cellular context.
  3. Impact on ferroptosis sensitivity:
    The compensatory NRF2 activation in TSC2(-) AML cells may provide an
    adaptive antioxidant responsesufficient to neutralize the pro-ferroptotic effects of FSP1 loss, thereby maintaining resistance to RSL3. In contrast, in ovarian and breast cancer cells where NRF2 is not constitutively active, FSP1 knockdown creates a more substantial redox vulnerability that tips cells toward ferroptosis.

We have added clarifying text in the Discussion section to highlight this point:
“Our results suggest that baseline NRF2 pathway activity in TSC2-null AML cells may render FSP1 dispensable for ferroptosis resistance. FSP1 knockdown in these cells induces a compensatory upregulation of NRF2 targets, potentially through redox imbalance or CoQ10/NAD(P)H disruption, which may counteract ferroptosis sensitization. This highlights a mechanistically distinct reliance on ferroptosis-resistance pathways across different cancer lineages.”

We acknowledge that further mechanistic dissection, for example, NRF2 reporter assays, KEAP1 oxidation studies, or metabolic flux analysis will be needed to fully characterize the NRF2–FSP1 compensatory axis and plan to pursue this in future work.

Comment 6: Considering the focus on iron metabolism, were direct assessments of intracellular iron concentrations or labile iron pools performed in all cell lines? Which specific iron transport proteins (e.g., ferroportin) or storage proteins (e.g., ferritin) were examined, and what were their expression levels and functional implications regarding ferroptosis resistance in each cell line?

Response 6: We agree the labile iron pool concentration is probably the most important aspect to experimentally determine, as it would give an indication of how iron metabolism is being modulated. This was carried out as well as a range of other metabolic markers that are linked to ferroptosis that included mitochondrial load and oxidative phosphorylation, lysosomal load and acidity. These experiments were not included in the original manuscript as it was considered less relevant, and the iron labile pool showed variation that we wished to pursue in a future paper to better understand resistant mechanisms. We have now added the data to the result section as Supplementary Figure 3:

‘We next used flow cytometry to assess the labile iron pool (LIP) and functional markers of mitochondria and lysosomes, as these metabolic parameters are closely linked to ferroptosis (Supplementary Figure S3). LIP was elevated in TSC2-deficient AML and ELT3 cells but was reduced in the Tsc2−/− MEFs. All three TSC2-deficient models ex-hibited higher mitochondrial load, consistent with mTORC1-driven mitochondrial bio-genesis via YY1 (10.1038/nature06322) and impaired mitophagy (10.1016/j.celrep.2016.09.054). Elevated mitochondrial load was associated with increased oxidative phosphorylation, as evidenced by JC-1 staining of mitochondrial membrane potential (ΔΨm) (10.1111/jcmm.15194). Lysosomal load was increased in the AML and ELT3 cells lacking TSC2, while lysosomal acidity was elevated in all TSC2-deficient lines, although this did not reach statistical significance in the ELT3 cells.

These alterations have potentially opposing effects on ferroptosis susceptibility. In-creased mitochondrial mass and oxidative phosphorylation are expected to elevate ROS, which can promote lipid peroxidation (https://doi.org/10.3389/fcell.2023.1226044). In the AML and ELT3 cells, the combination of high LIP and increased lysosomal acidity suggests enhanced ferritinophagy and iron release, potentially sensitizing cells to fer-roptosis. In contrast, the Tsc2−/− MEFs exhibited lower LIP despite high mitochondrial activity, suggesting preferential iron sequestration into iron–sulfur clusters, which may confer relative resistance unless iron availability is increased or GPX4 is inhibited. To-gether, these findings suggest that TSC2 loss can prime mitochondria for ROS generation and alter iron handling, with the ferroptotic outcome determined by the balance between iron sequestration and lysosomal iron release.’

Comment 7: The author could use the following papers (If applicable)
Identification of Novel Molecular Panel as Potential Biomarkers of PAN-Gastrointestinal Cancer
Screening: Bioinformatics and Experimental Analysis
The role of Sparasis Latifolia and Aerobic Exercise on Anti-inflammatory Activities and Regulation of Inflammatory Factors in the Intestine of Mice with Colon Cancer Under
Sparassis latifolia and exercise training as complementary medicine mitigated the 5-fluorouracil potent side effects in mice with colorectal cancer: bioinformatics approaches

Response 7: As we are already heavy with references (currently at 63) and our paper is not focused specifically on colon cancer (not examined, we felt it would not be applicable to also include these additional references).

Comment 8: Was a further experiment conducted to explicitly validate the radical scavenging action of necrostatin-1 at the concentrations employed in these cell lines? What were the outcomes?  What other studies could be conducted to distinguish between RIPK1 inhibition and radical scavenging as the principal mechanism underlying necrostatin-1's influence on RSL3-induced cell death in this context?

Response 8: During the initial review cycle with experimental corrections, we focused our attention on validation that ferroptosis was not inducing apoptosis (i.e., caspase activation). We also confirm that iron chelator and ferrostatin was sufficient to rescue cell death by ferroptosis induction and in multiple cell lines. Furthermore, we acknowledged that necrostatin-1 is known to have off-target antioxidant activity and cited these related manuscripts and highlighted that these could influence ferroptosis readouts. We feel that we have sufficiently addressed experimentally these control experiments to show that RSL3 is inducing an iron-dependent form of cell death that is in line with our current understanding.

To help researchers design more advanced methods in future work, we have now noted that our study used the classical necrostatin-1 analogue (as it was widely researched). To better distinguish between the mechanisms of antioxidant and RIPK1 inhibition in future work, there is a newer necrostatin-1 analogue termed Nec-1s that is likely to be more specific to inhibit RIPK1. We have added a note to the Discussion to reflect this point and clarify the limitations of interpreting necrostatin-1 in this context. We added this following to the discussion: ‘It should be noted that this study employed the classical necrostatin-1 analogue, which has been widely used in the literature but is also reported to possess off-target antioxidant activity that may influence ferroptosis readouts. While our results, together with iron chelation and ferrostatin rescue experiments, support the conclusion that RSL3 induces an iron-dependent form of cell death, future studies could employ the more specific necrostatin-1 analogue, Nec-1s, to better distinguish between effects mediated by RIPK1 inhibition and those arising from antioxidant activity.’

Reviewer 3 Report (New Reviewer)

Comments and Suggestions for Authors

The authors are addressing an important concern about ferroptosis resistance and can be relevant across multiple cancer models. It is also interesting that the authors identify a context dependent mechanisms of ferroptosis resistance because that is true and is important to identify which patients might benefit the most from the treatment. Overall the study is well done and will be of interest to the readers. Few concerns and suggestions to help strengthen the data:

  1. The authors have not shown how the mTOR pathway is affected specially in the cancer cells since they are not a clean system unlike the MEFs.
  2. The authors observe the rescue with Necrostatin and justify it but have they checked the necrosis markers with western blots? Also if the necrostatin rescues through regulation of oxidative stress then it might not as effective when the NRf2 inhibitor is used. The authors only use ferrostatin-1 for that experiment.
  3. Since the focus of the paper is on NRF2 inhibition as a strategy to overcome ferroptosis resistance, have the authors tried another NRF2 inhibitor besides the ML385?
  4. The authors see a striking difference in NRF2 distribution in the ELT3 system (-/+ Erastin) but would be good to show western blots for FSP1 and HMOX1 with the NRF2 inhibitor treatment to see how they are affected and similarly do an analysis of NRF2 in the MDA-MB-436 cells when they knockdown FSP1 to understand how to affect each other in a context dependent manner. Also, would have been good to try a different NRF2 inhibitor in the MDA-MB-436 to demonstrate convincingly that they do not respond to NRF2 inhibition.

Author Response

Comment 1: The authors are addressing an important concern about ferroptosis resistance and can be relevant across multiple cancer models. It is also interesting that the authors identify a context dependent mechanisms of ferroptosis resistance because that is true and is important to identify which patients might benefit the most from the treatment. Overall the study is well done and will be of interest to the readers. Few concerns and suggestions to help strengthen the data:

Response 1: Thank you for your kind comments. Prior to your comments, we had already went through an initial round of review with further experimentation that enhanced the study. We have now took on board your additional concerns and suggestions to strengthen the manuscript by including more detail of the cancer cell lines used and refined the result and discussion points.

Comment 2: The authors have not shown how the mTOR pathway is affected specially in the cancer cells since they are not a clean system unlike the MEFs.

Response 2: We agree that cancer cells are not as clean a system as the TSC2-deficient cell models used in the manuscript that show specific mTORC1 hyperactivity via loss of TSC2. The cancer cell lines used, PEO1, PEO4, OVCAR3, and MDA-MB-436 have known activation mutations within genes and changes to gene-expression that leads to enhanced mTORC1 activation. The genetic background of cell lines used was raised by reviewer 2 for the TSC2-deficient cell line models. Given this related point on the cancer cells used in this study, we opted to also include known mutations and gene-expression alterations within these 4 cancer cell lines that would enhance TSC2/mTORC1 signaling: ‘PEO1 and PEO4 are derived from the same patient. PEO1 is characterized by a homozygous BRCA2 mutation (c.5193C>G). BRAC2 mutations give rise to hyperactivity to AKT/TSC2/MTORC1 signaling (doi: 10.1038/onc.2010.603). PEO4 has a secondary mutation (Y1655Y) that restores BRCA2 function. OVCAR3 is characterized by mutation of TP53, and has increased PIK3CA transcription and reduced PTEN expression, which is known to enhance mTORC1 activation [ref doi: 10.1016/j.ygyno.2004.11.051]. MDA-MB-436 contains TP53, KRAS and PTEN mutations. For instance, the MDA-MB-436 have a known PTEN p.V85_splice mutation resulting in mTORC1 hyperactivity [ref doi: 10.1038/onc.2015.179].’

Comment 3: The authors observe the rescue with Necrostatin and justify it but have they checked the necrosis markers with western blots? Also if the necrostatin rescues through regulation of oxidative stress then it might not as effective when the NRf2 inhibitor is used. The authors only use ferrostatin-1 for that experiment.

Response 3l: While we justified the rescue of necrostatin, we now appreciate that we could have also looked at phosphorylation of RIP3 and MLKL as potential markers of necrosis (something we were not familiar with during the initial review and resubmission). We never considered the combination of necrostatin and use of a NRF2 inhibitor. Reviewer 2 also made an additional point regarding the use of Necrostatin, which we should address as a limitation of the study. We added this comment to the discussion to advance future research and to acknowledge the limitation of relying on necrostatin alone: ‘It should be noted that this study employed the classical necrostatin-1 analogue, which has been widely used in the literature but has also been reported to possess off-target antioxidant activity that may influence ferroptosis readouts [ref]. While our results, together with iron chelation and ferrostatin rescue experiments, support the conclusion that RSL3 induces an iron-dependent form of cell death, future studies could employ the more specific necrostatin-1 analogue, Nec-1s, to better distinguish between effects mediated by RIPK1 inhibition and those arising from antioxidant activity’.

Comment 4: Since the focus of the paper is on NRF2 inhibition as a strategy to overcome ferroptosis resistance, have the authors tried another NRF2 inhibitor besides the ML385?

Response 4: This valid point was also raised by reviewer 2, as side effects are common with most drug inhibitors. In our study, we explored NRF2 siRNA knockdown in the cancer cell models. We have added the following statement to the Discussion section, to acknowledge off-target effects:

‘We found that drug inhibition of NRF2 with ML385 resensitized TSC2-deficient cells to ferroptosis. While we use low concentrations of ML385 that are routinely employed to inhibit NRF2 activity, we acknowledge that off-target effects unrelated to NRF2 could potentially influence ferroptosis induction. However, the lack of cytotoxicity observed with ML385 alone, together with the corroborating data from NRF2 knockdown in cancer cell models, supports the specificity of our findings.’

Comment 5: The authors see a striking difference in NRF2 distribution in the ELT3 system (-/+ Erastin) but would be good to show western blots for FSP1 and HMOX1 with the NRF2 inhibitor treatment to see how they are affected and similarly do an analysis of NRF2 in the MDA-MB-436 cells when they knockdown FSP1 to understand how to affect each other in a context dependent manner. Also, would have been good to try a different NRF2 inhibitor in the MDA-MB-436 to demonstrate convincingly that they do not respond to NRF2 inhibition.

Response 5: We thank the reviewer for these insightful comments regarding NRF2 and additional experimental ideas to strengthen the manuscript. We are not in a viable position to carry out these extra experiments but agree that such experiments would be fruitful in a follow-up study that we are planning to work on. Our previous data with western blots or IF for NRF2 was not reliable for determining modest changes in NRF2 activity. We hope that the changes we observe in gene-expression alone indicating that there was an upregulation of NRF2-target genes upon FSP1 knockdown is sufficient to address this point, and sharing the data through a public repository would be useful for future related studies.

This comment is also related to reviewer 2’s comment, where we addressed the comment with the addition of this to the discussion: ‘Our results suggest that baseline NRF2 pathway activity in TSC2-null AML cells may render FSP1 dispensable for ferroptosis resistance. FSP1 knockdown in these cells induces a compensatory upregulation of NRF2 targets, potentially through redox imbalance or CoQ10/NAD(P)H disruption, which may counteract ferroptosis sensitization. This highlights a mechanistically distinct reliance on ferroptosis-resistance pathways across different cancer lineages.’

We acknowledge that further mechanistic dissection, for example, NRF2 reporter assays, KEAP1 oxidation studies, or metabolic flux analysis will be needed to fully characterize the NRF2–FSP1 compensatory axis and plan to pursue this in future work.

Round 2

Reviewer 2 Report (Previous Reviewer 4)

Comments and Suggestions for Authors

The author responded to the comments.

This manuscript is a resubmission of an earlier submission. The following is a list of the peer review reports and author responses from that submission.

Round 1

Reviewer 1 Report

Comments and Suggestions for Authors

Dear editor 

The manuscript entitled "Targeting NRF2 and FSP1 to Overcome Ferroptosis Resistance in TSC2-Deficient and Cancer Cells" describes the Ferroptosis Resistance inhibitions as a cancer therapies method through targeting NRF2 and FSP1. This manuscript can be considered for publication after major revision and addressing following comments point-by-ponits.

1- What is novely of your study? please compare your study (in introduction )with literature such as

FSP1, a novel KEAP1/NRF2 target gene regulating ferroptosis and radioresistance in lung cancers, Emmanuel et al. 10.18632/oncotarget.28301

FSP1 and NRF2 independently contribute to ferroptosis resistance in KEAP1 mutant non-small cell lung carcinoma,Kim et al., https://doi.org/10.21203/rs.3.rs-2921779/v1

2- Please prepare a graphical abstract to exactly present total protocol and mechanism of Ferroptosis Resistance inhibitions 

3- It is highly suggested to add two sections entitled " reagents" and "apparatus" to commercially introduce materials and systems. Moreover concisely explain application of them.

4- Conclusion is poorly written. Please complete it

5- Please increase quality of figures. The figures are merged so messy. They can be separated and magnify the legends.

Comments on the Quality of English Language

The english should be polished as well.

Reviewer 2 Report

Comments and Suggestions for Authors

Major comments:

  1. RNA-seq methods section is missing crucial processing information including Illumina flow cell, read length. “Reads were processed into clean data” line 181 is vague. Specific reference genomes used, gene annotation files and respective version numbers are missing. Finally, raw data (fastqs) need to be made publicly available using a data repository such as GEO. These are all crucial in order to ensure reproducibility.

  1. Crucial controls missing include assessment of TSC2 status in ELT3 cells. This is provided for MEFs and AML cells but appear in Figure 2, after cell viability experiments discussed in Figure 1 – these westerns should appear prior to these downstream experiments.

  1. Figure 1b shows cell viability data for ELT3 cells, yet downstream experiments focus on AML cells, with respect to gene expression. Are TSC2-null AML cells similarly less sensitive to Erastin and RSL3?

  1. A general theme regarding the use of inhibitors and other drugs to ask biological questions is the lack of any controls/PD markers that would indicate whether observed phenotypes are due to on-target effects (ferroptosis) or not. With RSL3 and Erastin treatment, markers such as GSG/GSSG ratio and apoptosis markers could be measured. Examples of why this is important are: In Figure 1c, necrostatin appears to rescue RSL3-mediated cell death – is this because of ferroptosis inhibition or necroptosis inhibition? It is also unclear for example, whether there is any apoptotic cell death occurring due to RSL3 treatment and whether Z-VAD-FMK at the used concentration would indeed working to inhibit caspase in these cells. Are these also observed upon Erastin treatment?

  1. The authors are missing a quantification for IF images and statistics (Figure 3c) to conclusively determine whether this is significantly different across TSC2-WT and TSC2-deficient cells. Are these also seen in MEFs and AML cells? The selective use of cell lines in some experiments and not others is not convincing. Is the NRF2 IF signal specific to NRF2 in the rat cells used here? It is unclear whether this antibody has been knock out validated in these cells by looking at manufacturer documentation – this needs to be addressed.

  1. The authors use ML385 co-treatment with RSL3 to conclude that NRF2 inhibition is sufficient to confer sensitivity to ferroptosis. As before, it is unclear whether ML385 treatment is on target, or whether the observed phenotype here is due to off-target toxicity. Is NRF2 really being selectively inhibited here? Can the co-treatment be rescued with ferrostatin and other approaches? The same concerns apply to FSP1 inhibition. Genetics is more specific than pharmacology, and as such redoing this experiment with genetic NRF2/FSP1 inhibition would be more convincing. This approach appears to however been taken in cancer cells in Figure 4 – although again whether this is ferroptosis dependent or independent remains unclear due to the lack of rescue experiments.

  1. The premise of inhibiting a secondary pathway to confer sensitivity to an drug as described in the TSC null and cancer cell line work would make sense when it can be leveraged to specifically target cancer cells and spare normal cells. As wild type TSC2 cells are already sensitive to RSL3 and erastin, and presumably are even more sensitive once NRF2 and/or FSP1 are inhibited, how would this co-treatment approach serve as a specific TSC-null cell killing mechanism? The same holds true for the cancer cells. There needs to be some discussion on this.

Minor comments:

  1. Missing citation in line 308 – ‘modulated by NRF2 during induction of ferroptosis’.

  1. Missing citations in lines 273-275 when describing use of inhibitors.

Reviewer 3 Report

Comments and Suggestions for Authors

Tasmia Tahsin et al. reported an interesting work about ferroptosis therapy enhancement in cancer. The topic well fit in the scope of Cancers. The manuscript was informative, and deserved for publication after a Minor Revision. Detailed comments as follows:

  1. In the Result subsection of Abstract, the critical data of expression fold-change should be provided with p
  2. In Figure 1C, will the addition of DFO as the ferroptosis inhibitor be better?
  3. In WB test, will the addition of xCT as a ferroptosis indicator be better?
  4. For CLSM images in Figure 3C, please add the semi-quantification analysis by Image J.

Reviewer 4 Report

Comments and Suggestions for Authors
  1. The bioinformatics is week.
  2.  Materials and Methods are unclear, especially the aimes and grouping.
  3. The manuscript mentions NRF2 and FSP1 as key regulators of ferroptosis resistance but fails to explore other potential pathways or factors that could influence this resistance. Why was there no discussion on the roles of key players like GPX4 or the impact of iron metabolism in greater detail?
  4. While the manuscript suggests patient stratification might improve therapeutic outcomes, there is no substantial data provided on how this stratification can be effectively implemented or what specific criteria should be used. What methodologies are suggested for determining patient eligibility for treatment targeting NRF2 and FSP1?
  5. The investigation primarily utilized a restricted range of cancer cell lines (ovarian and breast cancer). The applicability of the findings to a broader range of cancer types (e.g., lung or colorectal cancers) is not addressed. How might variations in ferroptosis resistance mechanisms differ in other cancer types?
  6. Can the authors clarify the sample sizes and controls used in cell viability and RNA sequencing experiments to ensure reproducibility?
  7. Could the authors provide a more balanced perspective by discussing alternative pathways that may also contribute to ferroptosis resistance alongside NRF2?
  8. How do the authors propose to establish a direct link between altered gene expression and functional outcomes in ferroptosis?
  9. The manuscript occasionally uses imprecise or inconsistent terminology (e.g., interchangeably using "ferroptosis" and "cell death" without clarifying the context). How will the authors ensure clarity and precision in the terminology used throughout the manuscript?
  10. The author could write the limitations and future studies in the discussion. The author could use the  https://doi.org/10.1186/s12935-024-03328-y and  https://doi.org/10.3390/biology12111426 (if applicable). Could the authors propose a clear roadmap for the next steps in research that would most effectively build on their findings?
  11. Clarify the term ferroptosis resistance.